# Incidence of lower limb deep vein thrombosis in patients with COVID-19 pneumonia through different *waves* of SARS-CoV-2 pandemic: A multicenter prospective study

Filippo Pieralli[1]*, Fulvio Pomero[2], Lorenzo Corbo[1], Alberto Fortini[3], Giulia Guazzini[1], Lisa Lastraioli[1], Fabio Luise[1], Antonio Mancini[4], Lucia Maddaluni[1], Alessandro Milia[1], Lucia Sammicheli[1], Filippo Mani[1], Rossella Marcucci[5]

**1** COVID-19 Intermediate Care Unit, Careggi University Hospital, Florence, Italy, **2** Internal Medicine COVID-19 Unit, Ospedale Michele and Pietro Ferrero, Verduno, Cuneo, Italy, **3** Internal Medicine COVID-19 Unit, San Giovanni di Dio Hospital, Florence, Italy, **4** Internal Medicine Unit, Ospedale del Valdarno, Montevarchi, Arezzo, Italy, **5** Department of Clinical and Experimental Medicine, University of Florence and Careggi Hospital, Florence, Italy

* filippopieralli@gmail.com

**Data Availability Statement:** An anonymized minimal data set has been uploaded in OSF (Open

## Abstract

### Objective

The aim of this study was to evaluate the incidence of deep vein thrombosis (DVT) of the lower limbs in patients hospitalized with COVID-19 pneumonia in a non-ICU setting according to the different *waves* of the SARS-CoV-2 pandemic.

### Methods

Multicenter, prospective study of patients with COVID-19 pneumonia admitted to Internal Medicine units in Italy during the *first (March-May 2020)* and *subsequent waves (November 2020 –April 2021)* of the pandemic using a serial compression ultrasound (CUS) surveillance to detect DVT of the lower limbs.

### Results

Three-hundred-sixty-three consecutive patients were enrolled. The pooled incidence of DVT was 8%: 13.5% in the *first wave*, and 4.2% in the *subsequent waves* (p = 0.002). The proportion of patients with early (< 4 days) detection of DVT was higher in patients during the *first wave* with respect to those of *subsequent waves* (8.1% vs 1.9%; p = 0.004). Patients enrolled in different waves had similar clinical characteristics, and thrombotic risk profile. Less patients during the *first wave* received intermediate/high dose anticoagulation with respect to those of the *subsequent waves* (40.5% vs 54.5%; p = 0.005); there was a significant difference in anticoagulant regimen and initiation of thromboprophylaxis at home (8.1% vs 25.1%; p<0.001).

Science Framework) repository accessible at
https://osf.io/mwd5n/.

**Funding:** The authors received no specific funding
for this work.

**Competing interests:** No authors have competing
interests

## Conclusions

In acutely ill patients with COVID-19 pneumonia, the incidence of DVT of the lower limbs showed a 3-fold decrease during the *first* with respect to the *subsequent waves* of the pandemic. A significant increase in thromboprophylaxis initiation prior to hospitalization, and the increase of the intensity of anticoagulation during hospitalization, likely, played a relevant role to explain this observation.

## 1. Introduction

The ongoing SARS-CoV-2 infection pandemic is generating a heavy burden of morbidity, mortality, and pressure for health-care systems all over the world. A well-documented feature of SARS-CoV-2 infection, especially in moderate to severe cases of COVID-19, is the significant prevalence of a coagulopathy [1, 2] characterized by elevated D-dimers and other bio humoral markers of activation of the coagulation cascade, with an increased incidence of arterial and venous thromboembolic events. Specifically, a high incidence of venous thromboembolism (VTE) has been reported in patients hospitalized for COVID-19 pneumonia. Indeed, the incidence of VTE in patients hospitalized for COVID-19 pneumonia in a non-ICU setting has been reported up to 20% [3–8]. In a recent large multicenter prospective study in non-ICU patients with moderate-severe COVID-19, we reported an incidence of deep vein thrombosis (DVT) of the lower limbs, using serial compression ultrasound (CUS) surveillance of 13.7% [9] during the first Italian wave of pandemic. These results have been recently confirmed by a systematic review and meta-analysis by Mansory et al, where people admitted at hospital for COVID-19 in a non-critical care setting showed a cumulative incidence of venous thromboembolism of 7.7% [10]. Nowadays, a general trend toward reduction of thrombotic events prevalence in patients hospitalized for COVID-19 pneumonia between the *first* and the *subsequent waves* of the SARS-CoV-2 pandemic has been described [11]. Similarly, a recently published paper by Katsoularis et al. describing the results of a large nationwide retrospective Swedish registry (the outcomes of VTE were defined by international classification of diseases, 9th and 10th revisions), confirmed the increased risk of VTE in patients with documented SARS-CoV-2 infection with respect to a matched cohort, with highest risk in patients with critical COVID-19 and during the first pandemic wave compared to other periods [12].

However, the confirmation and the magnitude of the suggested decrease in VTE risk over time, and the possible explanations of those findings, remains unclear. To explore this unanswered question, we compared the incidence of DVT of the lower limbs diagnosed by serial ultrasonographic surveillance in two cohorts of patients with COVID-19 admitted to Internal Medicine Units (IMUs) in two different pandemic periods between March 2020 and April 2021.

## 2. Patients, setting, and methods

This is a prospective cohort study carried out at two large Italian hospitals in two different periods [named *first (March-May 2020) and subsequent (November 20-April 21) waves*] of the SARS-CoV-2 pandemic. The COVID-19 Intermediate Care Unit of the Careggi University Hospital, the Internal Medicine COVID-19 Unit of the San Giovanni di Dio Hospital, both in Florence, and the Internal Medicine COVID-19 Unit). Patients were included in the study if they met all the following criteria: age ≥ 18 years; objectively confirmed diagnosis of

SARS-CoV-2 pneumonia obtained by real-time reverse transcription polymerase chain reaction (RT-PCR) assay on naso-pharyngeal swab and/or bronchoalveolar lavage; thoracic imaging (chest X-ray or CT scan) documenting pneumonia. Since, the observational nature of the study, no exclusion criteria, except for age below 18 years, were considered in the protocol.

All consecutive patients aged more than 18 years with a definitive diagnosis of SARS-CoV-2 pneumonia admitted to these units were enrolled in two different periods of time of the pandemic. Patients of the first cohort, were enrolled from March 21st to May 25th, 2020, during the *first wave* of the pandemic, while patients of the second cohort were enrolled during the *subsequent waves* from November 1st 2020 to April 30th 2021. On admission, thromboprophylaxis with enoxaparin or fondaparinux was prescribed to all patients according to protocols in use in hospitals at that time. Exceptions were the presence of absolute contraindications to anticoagulant therapy or indication to full dose anticoagulation. According to literature definitions [1] the doses of initial anticoagulation were defined as follows:

- Low dose anticoagulation equivalent to enoxaparin 20–40 mg or fondaparinux 1.5–2.5 mg per day.

- Intermediate dose anticoagulation equivalent to enoxaparin 60–80 mg per day.

- High dose anticoagulation equivalent to enoxaparin 120–160 mg or fondaparinux 5 to 10 mg per day, or oral anticoagulation with vitamin K antagonists (VKA) or direct oral anticoagulants (DOACs).

All patients were screened for DVT of the lower limbs by a surveillance protocol with serial color-coded Doppler and compression ultrasonography (CUS) within 72 hours since admission and subsequently at 5–7-day intervals, and before discharge. The complete ultrasound protocol has been previously described in detail and published elsewhere [9]. Briefly, the ultrasound scan was performed by experienced physicians along the proximal femoral and popliteal district bilaterally, by a three-point examination of the common and superficial femoral veins, and the popliteal veins [13]. The examination of the infrapopliteal vein district was not mandatory and was only performed at discretion of the examining physician. The lack of vein compression was the only diagnostic criterion for the diagnosis of DVT. Each time DVT was diagnosed, patients were finally shifted to full dose anticoagulation. DVTs were considered asymptomatic or symptomatic in relation to the absence or presence of clinical signs (i.e. leg swelling, pain, or both) suggesting a venous thrombosis. A multidetector computed tomographic pulmonary angiography (CTPA) was ordered upon clinical judgement in those cases with clinical suspicion of pulmonary embolism (PE), and no predefined protocol was in place for ordering CTPA.

Demographic and clinical features, as well as laboratory variables were recorded. Blood cell count, aPTT, PT/INR, fibrinogen, C-reactive protein (CRP), and D-dimers (expressed as ng/mL FEU–fibrinogen equivalent unit) were obtained by blood samples collected on admission and at 48 to 72 hours intervals. The method and the laboratory normal values of D-dimers were the same for all the two hospitals' laboratories with an upper reference limit of 500 ng/mL FEU. We defined "peak D-dimers" as the highest value during hospital stay for patients without DVT and the highest value at the time (±24h) DVT was diagnosed. The Padua Prediction Score (PPS) [14] was calculated for each patient at admission, the results are expressed on a 0 to 20 points scale, and a score greater or equal to 4 points defines patients at high risk of VTE. All major hemorrhagic complications were recorded; accordingly, they were defined as major if leading to hemodynamic instability, need for blood transfusions, or occurring at any critical site, and/or leading to death.

The primary end-point of the study was the incident diagnosis (cumulative incidence) of DVT of the lower limbs objectively confirmed by CUS and the comparison of the cumulative incidence between the *first* and the *subsequent waves* of COVID-19 pandemic.

The study protocol was approved by the ethics committee of the coordinating center, Azienda Ospedaliera Universitaria Careggi, Florence, Italy (COCORA protocol 17104), and was performed in agreement with the principles set in the Declaration of Helsinki for studies involving human beings. Informed consent was not needed, since point-of-care ultrasonography is a standard of care for the evaluation and monitoring of patients with COVID-19; informed consent was waived by the ethics committee, only signed consent for personal data collection and treatment was requested, and all data were collected and analyzed anonymously.

## 3. Statistical analysis

Continuous variables were expressed as mean ± standard deviation or as median and inter-quartile ranges (IQR; 25th–75th percentile) as appropriate. In general, statistical comparisons were performed using Student's t test and one-way ANOVA models for the comparison of continuous normally distributed variables and Mann-Whitney *U* test for continuous not normally distributed variables. The Chi-square test or Fisher's exact test were used for the comparison of categorical variables. All p values were two-tailed and considered significant when <0.05 (95% CI). Analyses and chart designs were performed using Statistical Package for Social Sciences software 21.0 (SPSS, Chicago, Illinois, USA).

## 4. Results

### Overall population characteristics

Three-hundred-sixty-three was the global number of patients enrolled in the two cohorts of the study (278 patients at Careggi Hospital, and 85 patients at S. Giovanni di Dio Hospital, respectively). General clinical, demographic characteristics and laboratory findings of the entire population are described in Table 1. The proportion of male and elderly patients was predominant (male 63.1%; 51,7% with age > 70 years), and there was a relevant prevalence of cardiovascular diseases and diabetes. All patients had a Padua Prediction Score (PPS) ≥ 5, featuring a high risk of VTE in all cases; specifically 21,5% had a score > 6 points. All patients had moderate-severe pneumonia with respiratory failure requiring standard oxygen therapy, and/or high flow oxygenation with nasal cannulas and/or non invasive mechanical ventilation by helmet or mask. All patients received CUS of proximal veins as per protocol, and ultrasound evaluation was extended to the distal infrapopliteal venous district in 170 patients (46.8%). The overall incidence of DVT by CUS surveillance during hospitalization was 8% for the entire cohort. In 15 (4.1%) patients DVT was proximal, while in 14 (3.9%) patients an infrapopliteal DVT was diagnosed; four patients had bilateral DVT, and in two DVT involved both the proximal and distal district. Notably, most patients (86.2%) had asymptomatic DVT, and only 4 patients were symptomatic (leg swelling and/or pain): 2 popliteal and 2 femoral DVTs.

The detection of DVT occurred early in the course of the hospitalization. Specifically, the timing of DVT detection from admission was within 4 days in 16 patients (55.2%), between day 5 and 10 in 12 patients (41.4%) (Fig 1), and only 1 patient has been diagnosed beyond 10 days since admission. Overall, 20 patients (5,5%) were diagnosed with PE at CTPA; of these, 6 had a concomitant DVT of the lower limbs, while in 14 cases PE was isolated.

Nearly all patients (99.2%) received anticoagulation, being enoxaparin the most frequently administered anticoagulant drug (89.8%), while only a minority of subjects received fondaparinux (1.9%), or VKA/DOACs (6.6%) for concomitant clinical indications to receive full-dose anticoagulation. Anticoagulation at various intensity was already in place at home prior to

**Table 1. Demographic, clinical characteristics and laboratory findings and outcomes of the pooled cohorts of patients.**

|  | Overall population |
|---|---|
| Age years (mean±SD) | 70 ± 13 |
| Age> 70 years, n (%) | 188 (51.7) |
| Female gender, n (%) | 144 (36.9) |
| **Comorbidities** |  |
| Hypertension, n (%) | 219 (60.3) |
| Cardiovascular disease, n (%) | 134 (36.9) |
| Diabetes, n (%) | 77 (21.2) |
| COPD, n (%) | 64 (17.6) |
| Obesity (BMI>30 Kg/m$^2$), n (%) | 61 (16.8) |
| Active cancer, n (%) | 28 (7.7) |
| **Main laboratory findings** |  |
| Platelet count, 10$^9$/L (mean±SD) | 198000 ± 93833 |
| White Blood Cell, 10$^9$/L (mean±SD) | 7.38 ± 4.48 |
| Creatinine, mg/dL (mean±SD) | 1.00 ± 1.14 |
| PT, activity % (mean±SD) | 74.9 ± 16.0 |
| aPTT, seconds (mean±SD) | 31.2 ± 7.3 |
| Fibrinogen, mg/dL (mean±SD) | 584.3 ± 194.4 |
| C reactive protein, mg/L, (median—IQR) | 74 (32–136) |
| D-dimer peak value, ng/mL (median—IQR) | 2188 (1109–7110) |
| **Padua Prediction Score** |  |
| 5, n (%) | 106 (29.2) |
| 6, n (%) | 179 (49.3) |
| >6, n (%) | 78 (21.5) |
| Mean±SD | 6.15±1.3 |
| **Diagnosis of DVT** |  |
| Total, n (%) | 29 (8.0) |
| Proximal, n (%) | 15 (4.1) |
| Distal, n (%) | 14 (3.9) |
| **Timing of DVT diagnosis since admission** |  |
| Day 0–4, n (%) | 16 (55.2%) |
| Day 5–10, n (%) | 12 (41.4%) |
| Day >10, n (%) | 1 (3.4%) |
| **Outcome measures** |  |
| In-hospital length of stay, days (mean±SD) | 19.7 ± 10.7 |
| In-hospital death, n (%) | 91 (25.1) |

COPD: Chronic obstructive pulmonary disease; PT: prothrombin time; aPTT: activated partial thromboplastin time; DVT: deep vein thrombosis; ICU: Intensive Care Unit.

hospitalization for COVID pneumonia in 66 patients (18.2%), with a significantly lower prevalence in the first wave (8.1% vs 25,1%; p<0.001).

The overall incidence of major bleedings was low across the entire cohort, and only four patients experienced major hemorrhagic complications [2 in the *first wave (1.3%)* and 2 in the *subsequent (0.9%) waves*). One patient developed a large retroperitoneal psoas muscle hematoma requiring blood transfusions and drainage on high-dose anticoagulation; three patients experienced gastroduodenal bleeding from gastric (2 patients) and duodenal ulcers (1 patient) requiring endoscopic treatment and blood transfusions.

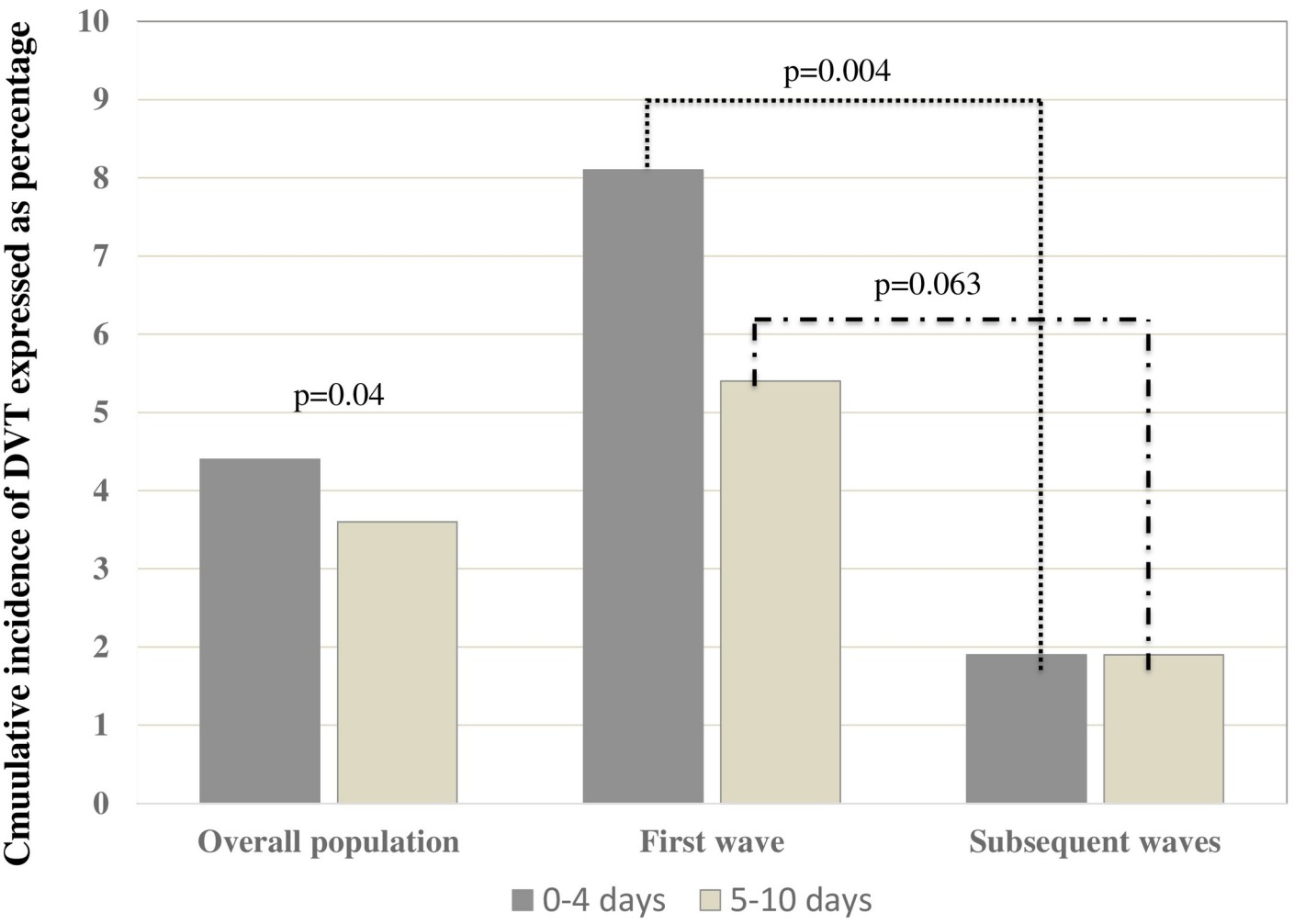

Figure 1. Cumulative incidence of DVT in two waves of the COVID-19 pandemic according to time since hospital admission.

Fig 1. Cumulative incidence of DVT in two waves of the COVID-19 pandemic according to time since hospital admission.

### Comparison of the characteristics of the population during the *first* and *subsequent waves* of the pandemic

The patients enrolled in the *first* and *subsequent waves* cohorts were 148 and 215, respectively. In general, they shared similar characteristics (Table 2); of note, patients observed in the *first wave* were slightly older and less obese than those observed in the *subsequent waves*. No significant difference was observed in median "D-dimer peak" values between groups (1685 [IQR 1045–5682] ng/mL vs 2342 [IQR 1224–10743] ng/mL; p = 0.176). The cumulative incidence of DVT was significantly higher in the *first wave* with respect to that of the *subsequent waves* (13.5% vs 4.2%; p = 0.002), this difference was maintained even when the proximal and distal district were considered (Table 2). In general, DVT was diagnosed early in the course of hospitalization; however, a significant difference in the timing of DVT diagnosis was recorded in the cohorts of patients observed during the *first* and the *subsequent waves* of the COVID-19

**Table 2. Demographic, clinical characteristics and laboratory findings and outcomes of patients with e without DVT.**

| | *First wave* | *Subsequent waves* | *p* |
|---|---|---|---|
| | *(March-May 2020)* | *(November 2020- April 2021)* | |
| N | 148 | 215 | |
| Age, years (mean±SD) | 72 ± 13 | 69 ± 14 | 0.033 |
| Female gender, n (%) | 62 (41.9) | 82 (38.1) | 0.513 |
| **Clinical features** | | | |
| Hypertension, n (%) | 84 (56.8) | 135 (62.8) | 0.275 |
| Cardiovascular disease, n (%) | 50 (33.8) | 84 (39.1) | 0.321 |
| Diabetes, n (%) | 32 (21.6) | 45 (20.9) | 0.897 |
| COPD, n (%) | 24 (16.2) | 40 (18.6) | 0.578 |
| BMI>30 Kg/m$^2$, n (%) | 14 (9.5%) | 47 (21.9%) | 0.002 |
| Active cancer, n (%) | 12 (8.1) | 16 (7.4) | 0.843 |
| **Main laboratory findings** | | | |
| Platelet count, 109/L (mean±SD) | 220567 ± 96698 | 211855 ± 91868 | 0.385 |
| White Blood Cell, 10$^9$/L (mean±SD) | 7.80 ± 4.01 | 8.74 ± 4.75 | 0.051 |
| Creatinine, mg/dL (mean±SD) | 1.45 ± 1.61 | 1.10 ± 0.62 | 0.003 |
| PT, activity % (mean±SD) | 72.29 ± 15.47 | 76.70 ± 16.07 | 0.101 |
| aPTT, seconds (mean±SD) | 30.75 ± 6.22 | 31.46 ± 7.94 | 0.369 |
| Fibrinogen, mg/dL (mean±SD) | 581.86 ± 151.55 | 586.00 ± 219.90 | 0.843 |
| C reactive protein, mg/L, (median-IQR) | 76.5 (32.8–170) | 68 (31–121) | 0.401 |
| D-dimer peak value, ng/mL (median-IQR) | 1685 (1045–5682) | 2342 (1224–10743) | 0.176 |
| **Padua Prediction Score** | | | |
| Points (mean±SD) | 6.14 ± 1.23 | 6.31 ± 1.40 | 0.155 |
| **Intensity of anticoagulation** | | | |
| Low dose, n (%) | 88 (59.5) | 95(44.2) | 0.005 |
| Intermediate dose, n (%) | 29 (19.6) | 64 (29.8) | 0.037 |
| High dose, n (%) | 31 (20.9) | 53 (24.7) | 0.449 |
| Thromboprophylaxis at home, n (%) | 12 (8.1) | 54 (25.1) | < 0.001 |
| Deep venous thrombosis, n (%) | 20 (13.5) | 9 (4.2) | 0.002 |
| Proximal, n (%) | 10 (6.8) | 5 (2.3) | 0.037 |
| Distal, n (%) | 10 (6.8) | 4 (1.9) | 0.025 |

DVT: deep vein thrombosis; COPD: Chronic obstructive pulmonary disease; PT: prothrombin time; aPTT: activated partial thromboplastin time; ICU: Intensive Care Unit.

pandemic (Fig 1). In general, the proportion of patients with detection of DVT within 4 days since hospital admission was higher in patients observed during the *first wave* of the pandemic with respect to those of the *subsequent waves* ((n = 12; 8.1% vs n = 4; 1.9%; p = 0.004); while no difference was observed in those patients diagnosed with DVT after day 5 since admission (n = 8, 5.4% vs n = 4, 1.9%; p = 0.063) (Fig 1). Patients observed in the *second waves* received more intermediate and less low-dose anticoagulation with respect to those enrolled in the *first wave* (Table 2); namely, the proportion of patients receiving different intensity initial anticoagulant treatment in the *first* and *second waves* were as follows: low dose (thromboprophylaxis) 59.5% vs 44.2% (p<0.005), intermediate dose 19.6% vs 29.8% (p = 0.037), and high dose 20.9% vs 24.7% (p = 0.449) (Table 2).

Notably, nearly a threefold increase in the proportion of patients receiving anticoagulant prophylaxis at home was observed in the *second wave* with respect to those of the *first wave* (25.1% vs 8.1%; p<0.001).

## 5. Discussion

In this multicenter study, a protocol with systematic ultrasonography for DVT surveillance of the lower limbs in patients admitted to IMUs with COVID-19 pneumonia detected a cumulative incidence of 8% in the COVID-19 pandemic period included between March 2020 and April 2021. The proportion of diagnosed thromboses was equally distributed between proximal and distal DVTs. According to the different periods of the pandemic, we observed nearly a three-fold decrease in the cumulative incidence of DVT in the later *waves* (November 2020-April 2021) with respect to the early *wave* (March-May 2020) of COVID-19 pandemic (4.2% vs 13.5%; p = 0.002). There is a widespread perception of a trend toward the reduction in prevalence of venous thromboembolism in hospitalized patients with COVID-19, but this observation has been supported mainly by direct clinical observation and perception, while sound data are inconclusive [15]. Ad hoc studies evaluating prospectively the temporal trend of DVT incidence in patients hospitalized for COVID-19 are lacking, and we can only extrapolate data by indirect comparison of studies that were carried out at different times of the pandemic. Pooled data from a metanalysis published by Zhang et al. during the period March-November 2020 (first *wave*) showed a cumulative incidence of DVT of 7% in patients hospitalized for COVID-19 outside the ICU [16]; while, two recent studies pertaining to the second phases of the pandemic reported a cumulative incidence of DVT in the range of 1–3% in non-ICU patients, features similar to 4.2% incidence found in the second *wave* in our study [10, 17–19].

The largest prospective, and more recently published study to date, is a Dutch multicenter study that evaluated the incidence of thrombotic cardiovascular complications during the first and second *wave* of COVID-19 pandemic in a heterogeneous population of persons hospitalized for COVID-19 in the ICU e non-ICU setting [11]. The overall incidence of thrombotic complications included in the primary composite end-point (VTE, DVT, myocardial infarction, stroke and systemic arterial embolism) declined over time during the pandemic. However, no predefined VTE screening strategy was used in that study, and diagnostic tests were applied only in patients with clinically suspected thrombotic complications, including DVT, without a standardized approach. Moreover, while the cumulative incidence of overall thrombotic complications declined over time, the cumulative incidence of VTE did not, particularly in non ICU patients [11]. Recently, the results of a large Swedish registry confirmed that people infected by SARS-CoV-2 are generally at increased risk of VTE with respect to matched non infected control patients; the risk was higher during the *first wave* of the pandemic with respect to the *second* and *third waves* [12].

To the best of our knowledge this is the first *ad hoc* prospective study, directly comparing the incidence of DVT with a dedicated surveillance protocol during the earlier and later phases of COVID-19 pandemic, in which an objective documentation of a significant decrease in cumulative incidence of DVT was reported, confirming the generally perceived reduction in the incidence of DVT over the time of COVID-19 pandemic.

How can we explain a three-fold decrease in the cumulative incidence of DVT in patients hospitalized for COVID-19 during the different *waves* of the pandemic?

The exact reasons of this observation are still unclear, and likely there are several explanations for this finding. Many aspects of COVID-19 pathophysiology and treatment and prevention are possible causative factors. Among others, the difference in demographic and clinical characteristics of patients, changes in management strategies, including different anticoagulant regimens, and vaccination, and possibly the different impact of SARS-CoV-2 variants on coagulopathy, can play a significant role [20–22]. It is well known that SARS-CoV-2 through the binding to ACE2 receptor can trigger direct vascular injury, and may affect signaling pathways,

leading to acute cardiovascular injury [23]. Similarly, the acute systemic inflammatory response syndrome induced by SARS-CoV-2 infection, caused by a dysregulated reaction in pro-inflammatory cytokines release and production (known as cytokines storm), determines endothelial damage with cardiovascular consequences, such as venous thrombosis and cardiac injury [23]. Nevertheless, what is the role of the different SARS-CoV-2 variants and the effects of vaccination in the determination of different vascular injury burden is currently unknown.

Specifically, in this study, there were no significant differences in major clinical risk factors for venous thromboembolism as documented by similar risk profile by PADUA score in the two different study periods. Thus, the absence of difference of VTE risk profile as evaluated by a comprehensive instrumentt such as the PADUA score, when considering both median PADUA scores and risk class representation, cannot explain the findings of the significantly reduced incidence of DVT by ultrasound surveillance over time.

Additionally, more patients in the later *waves* received higher intensity anticoagulation with respect to those in the first *wave*, and a significant difference in thromboprophylaxis initiated at home before hospitalization was observed during the different *waves* of the pandemic. The proportion of patients that received thromboprophylaxis prior to hospitalization was roughly three times higher in the later *waves* than in the first *wave* (25.1% vs 8.1%; p<0.001). Therefore, the modification of anticoagulation strategies during the pandemic could be a major determinant of the observed reduction of DVT incidence through different *waves*. The change in management reflects the evolving strategies of COVID-19 treatment over time, even not always supported by sound evidence [18]. In fact, the clinical benefits in initiating thromboprophylaxis early at home in patients with SARS-CoV-2 associated mild-to-moderate pneumonia has not been established, and current guidelines advice not starting anticoagulation for acutely ill COVID-19 outpatients with mild-to-moderate disease not requiring hospitalization, based mainly on the results of ACTIV-4b trial [24–27]. Nevertheless in Italy, in order to reduce the burden on the hospital system, many patients with moderate COVID-19 were followed at home by dedicated COVID-19 teams and treated with low flow oxygen support, low dose steroids, and thrombophylaxis in a very similar fashion to those patients admitted to a non-critical care hospital setting. The finding that most of the reduction in DVT incidence between the two *waves* occurred in the early phase of hospitalization (first 4 days since admission), seems to support a possible protective role of at home anticoagulation treatment.

This study has some limitations. First, the 3-point CUS protocol was intended to detect proximal DVT and did not include the extension of US to whole leg; as previously described, [9] this choice was to obtain reliable data that could have general impact, since in most IMUs there is a staff trained and expert in performing CUS of proximal venous district, while the examination of the infrapopliteal veins require more expertise and is to feasible widely. Nonetheless, an underestimation of DVT incidence of the distal district cannot be ruled out. Second, we cannot exclude that the lack of standardization of anticoagulant doses, caused by the evolving strategies over time and the absence of definitive evidence [24, 25] could have had some impact on DVT incidence.

In conclusion, the findings of this *ad hoc* multicenter study confirm the reduction of DVT incidence over time during the *first* and later *waves* of COVID-19 pandemic in patients hospitalized outside the ICU, screened through a surveillance protocol by serial CUS of the lower limbs. These findings need further confirmation in different realities and setting to appreciate the magnitude of this observations.

## Author Contributions

**Conceptualization:** Filippo Pieralli, Fulvio Pomero, Rossella Marcucci.

**Data curation:** Filippo Pieralli, Lorenzo Corbo, Giulia Guazzini, Lisa Lastraioli, Fabio Luise, Lucia Sammicheli.

**Formal analysis:** Filippo Pieralli, Fulvio Pomero, Alberto Fortini.

**Investigation:** Antonio Mancini, Lucia Maddaluni, Alessandro Milia, Lucia Sammicheli, Filippo Mani.

**Methodology:** Rossella Marcucci.

**Supervision:** Alberto Fortini, Lisa Lastraioli, Antonio Mancini.

**Validation:** Alessandro Milia.

**Writing – original draft:** Filippo Pieralli, Fulvio Pomero, Lorenzo Corbo.

**Writing – review & editing:** Filippo Pieralli, Fulvio Pomero, Lorenzo Corbo, Alberto Fortini, Giulia Guazzini, Fabio Luise, Filippo Mani, Rossella Marcucci.

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
