## [Decision Letter · Decision Letter 0]

14 Nov 2022

PONE-D-22-24035

Incidence of lower limb deep vein thrombosis in patients with COVID-19 pneumonia through different waves of SARS-CoV-2 pandemic: a multicenter prospective study.

PLOS ONE

Dear Dr. Pieralli,

Thank you for submitting your manuscript to PLOS ONE. After careful consideration, we feel that it has merit but does not fully meet PLOS ONE’s publication criteria as it currently stands. Therefore, we invite you to submit a revised version of the manuscript that addresses the points raised during the review process.

We look forward to receiving your revised manuscript.

Kind regards,

Chiara Lazzeri

Academic Editor

PLOS ONE

Journal Requirements:

Pieralli F, Pomero F, Giampieri M, Marcucci R, Prisco D, Luise F, et al. (2021) Incidence of deep vein thrombosis through an ultrasound surveillance protocol in patients with COVID-19 pneumonia in non-ICU setting: A multicenter prospective study. PLoS ONE 16(5): e0251966. https://doi.org/10.1371/journal.pone.0251966

In your revision ensure you cite all your sources (including your own works), and quote or rephrase any duplicated text outside the methods section. Further consideration is dependent on these concerns being addressed.

“NO - No authors received fundings for this work”

Reviewers' comments:

Reviewer's Responses to Questions

**Comments to the Author**

1. Is the manuscript technically sound, and do the data support the conclusions?

Reviewer #1: Yes

Reviewer #2: Yes

2. Has the statistical analysis been performed appropriately and rigorously? 

Reviewer #1: Yes

Reviewer #2: Yes

3. Have the authors made all data underlying the findings in their manuscript fully available?

Reviewer #1: Yes

Reviewer #2: Yes

4. Is the manuscript presented in an intelligible fashion and written in standard English?

Reviewer #1: Yes

Reviewer #2: Yes

5. Review Comments to the Author

Reviewer #1: The paper presents a high quality of processing in terms of methodology, contents and results. As regards the introduction, it would be appropriate to increase the number of references and the length of the introduction, also introducing the background paragraph to provide a broader overview of the existing scientific literature also through a rapid revision of the same.

From a methodological point of view, I recommend providing the inclusion and exclusion criteria for recruited subjects more explicit because it would give further added value to this high-quality work.

The results are well represented both graphically and from a descriptive point of view, however, English should be improved.

The discussion is well structured, but I would suggest to increase the references because they are very few compared to the contents reported

Reviewer #2: The authors aimed to evaluate the incidence of deep vein thrombosis (DVT) of the lower limbs in patients hospitalized with COVID-19 pneumonia in a non-ICU setting according to the different waves of the SARS-CoV-2 pandemic.

The topics is relevant. Overall considered the paper is well written but the discussion needs to be a little expanded including more detailed pathophysiological considerations between cardiovascular diseases and COVID-19. For example cite and comment the article by Ielapi N et al. Cardiovascular disease as a biomarker for an increased risk of COVID-19 infection and related poor prognosis. Biomark Med. 2020 Jun;14(9):713-716. doi: 10.2217/bmm-2020-0201.

6. PLOS authors have the option to publish the peer review history of their article (what does this mean?). If published, this will include your full peer review and any attached files.

Reviewer #1: No

Reviewer #2: No

---

## [Author Response · Author response to Decision Letter 0]

18 Dec 2022

Dear Editor, we are grateful for the opportunity to resubmit our paper entitled “Incidence of lower limb deep vein thrombosis in patients with COVID-19 pneumonia through different waves of SARS-CoV-2 pandemic: a multicenter prospective study.”

Below you will find the complete rebuttal to the comments and queries posed by the reviewers.

We hope you will now find the manuscript suitable for consideration of publication in your Journal. 

Filippo Pieralli and Fulvio Pomero on the behalf of all coauthors.

Journal requirements.

We noticed you have some minor occurrence of overlapping text with the following previous publication(s), which needs to be addressed:

Pieralli F, Pomero F, Giampieri M, Marcucci R, Prisco D, Luise F, et al. (2021) Incidence of deep vein thrombosis through an ultrasound surveillance protocol in patients with COVID-19 pneumonia in non-ICU setting: A multicenter prospective study. PLoS ONE 16(5): e0251966. https://doi.org/10.1371/journal.pone.0251966

In your revision ensure you cite all your sources (including your own works), and quote or rephrase any duplicated text outside the methods section. Further consideration is dependent on these concerns being addressed.

The overlapping text with our previous work, Pieralli F, Pomero F, Giampieri M, Marcucci R, Prisco D, Luise F, et al. (2021) Incidence of deep vein thrombosis through an ultrasound surveillance protocol in patients with COVID-19 pneumonia in non-ICU setting: A multicenter prospective study. PLoS ONE 16(5): e0251966. https://doi.org/10.1371/journal.pone.0251966, has been rephrased weather needed. 

Please provide additional details regarding participant consent. In the ethics statement in the Methods and online submission information, please ensure that you have specified what type you obtained (for instance, written or verbal, and if verbal, how it was documented and witnessed). If your study included minors, state whether you obtained consent from parents or guardians. If the need for consent was waived by the ethics committee, please include this information.

The study protocol was approved by the ethics committee of the coordinating center, Azienda Ospedaliera Universitaria Careggi, Florence, Italy (COCORA protocol 17104). Informed consent was not needed, since point-of-care ultrasonography is a standard of care for the evaluation and monitoring of patients with COVID-19; only signed consent for personal data collection and treatment was requested; whenever the collection of the written consent for personal data collection was not feasible, a verbal consent witnessed by healthcare bystanders was obtained. All data were collected and analyzed anonymously.

Thank you for stating the following financial disclosure:

“NO - No authors received fundings for this work”.

The following statement has been now included at the end of the text.

None of the authors received specific funding for this work.

In your Data Availability statement, you have not specified where the minimal data set underlying the results described in your manuscript can be found. PLOS defines a study's minimal data set as the underlying data used to reach the conclusions drawn in the manuscript and any additional data required to replicate the reported study findings in their entirety. All PLOS journals require that the minimal data set be made fully available. For more information about our data policy, please see http://journals.plos.org/plosone/s/data-availability.

A minimal data set has been uploaded in OSF (Open Science Framework) repository accessible at https://osf.io/mwd5n/

Filippo Pieralli

---

## [Editor Report · Decision Letter 1]

26 Dec 2022

Incidence of lower limb deep vein thrombosis in patients with COVID-19 pneumonia through different waves of SARS-CoV-2 pandemic: a multicenter prospective study.

PONE-D-22-24035R1

Dear Dr. Pieralli,

We’re pleased to inform you that your manuscript has been judged scientifically suitable for publication and will be formally accepted for publication once it meets all outstanding technical requirements.

Kind regards,

Chiara Lazzeri

Academic Editor

PLOS ONE
---

## [Editor Report · Acceptance letter]

3 Jan 2023

PONE-D-22-24035R1 

Incidence of lower limb deep vein thrombosis in patients with COVID-19 pneumonia through different waves of SARS-CoV-2 pandemic: a multicenter prospective study. 

Dear Dr. Pieralli:

I'm pleased to inform you that your manuscript has been deemed suitable for publication in PLOS ONE. Congratulations! Your manuscript is now with our production department. 

Kind regards, 

on behalf of

Dr. Chiara Lazzeri 

Academic Editor

PLOS ONE